# MiR-106a-5p by Targeting *MAP3K2* Promotes Repair of Oxidative Stress Damage to the Intestinal Barrier in Prelaying Ducks

**DOI:** 10.3390/ani14071037

**Published:** 2024-03-28

**Authors:** Li Zhang, Xiang Luo, Rui Tang, Yan Wu, Zhenhua Liang, Jingbo Liu, Jinsong Pi, Hao Zhang

**Affiliations:** 1Hubei Key Laboratory of Animal Embryo Engineering and Molecular Breeding, Institute of Animal Husbandry and Veterinary Science, Hubei Academy of Agricultural Sciences, Wuhan 430064, China; zl9588623@163.com (L.Z.); luoxiangcsq@163.com (X.L.); q1207515167@163.com (R.T.); wuyanwh@163.com (Y.W.); liangzhenghua2046@163.com (Z.L.); pijinsong@sina.com (J.P.); 2School of Life Science and Engineering, Southwest University of Science and Technology, Mianyang 621010, China; liuswust@163.com

**Keywords:** apla-miR-106a-5p, intestinal barrier function, MAP3K2, oxidative stress, prelaying ducks

## Abstract

**Simple Summary:**

During the transition of ducks from floor to cage rearing, their intestinal barrier function is severely disrupted, adversely affecting economic performance. This study investigated the role of a crucial class of small noncoding RNAs, known as microRNAs (miRNAs), in repairing oxidative stress-induced damage to the duck intestinal barrier. In this study, we observed a significant downregulation of apla-miR-106a-5p during the repair of intestinal barrier damage. Additionally, we found that apla-miR-106a-5p can target and bind to *MAP3K2*, exerting post-transcriptional regulation and promoting the repair of intestinal barrier damage. These findings contribute valuable insights into the mechanisms involved in the repair of oxidative stress-induced damage to the intestinal barrier in ducks.

**Abstract:**

Under caged stress conditions, severe disruptions in duck intestinal barrier function, which adversely affect economic performance, have been observed. MiRNAs play a crucial role in cellular processes, but the mechanisms underlying their involvement in repairing oxidative stress-induced damage to duck intestinal barriers have not been elucidated. We performed miRNA-seq and protein tandem mass tagging (TMT) sequencing and identified differentially expressed miRNAs and proteins in oxidative stress-treated ducks. Dual-luciferase reporter vector experiments, RT-qPCR, and Western blotting revealed the regulatory role of apla-miR-106a-5p/MAP3K2 in intestinal barrier damage repair. The results showed that oxidative stress led to shortened villi and deepened crypts, impairing intestinal immune function. Significant downregulation of apla-miR-106a-5p was revealed by miRNA-seq, and the inhibition of its expression not only enhanced cell viability but also improved intestinal barrier function. TMT protein sequencing revealed MAP3K2 upregulation in caged-stressed duck intestines, and software analysis confirmed *MAP3K2* as the target gene of apla-miR-106a-5p. Dual-fluorescence reporter gene experiments demonstrated direct targeting of *MAP3K2* by apla-miR-106a-5p. RT-qPCR showed no effect on *MAP3K2* expression, while Western blot analysis indicated that MAP3K2 protein expression was suppressed. In summary, apla-miR-106a-5p targets *MAP3K2*, regulating gene expression at the transcriptional level and facilitating effective repair of intestinal barrier damage. This discovery provides new insights into the molecular mechanisms of physiological damage in ducks under caged stress, offering valuable guidance for related research.

## 1. Introduction

Intensive and facility-based breeding of laying ducks is a novel farming approach that offers several advantages, including mitigating pollution in the laying duck industry, increasing biological safety and product quality, and boosting the efficiency of laying duck production through standardized practices [1,2]. The typical age for transitioning laying ducks into a cage-rearing system is approximately 90 ± 10 days. However, the shift from ground-based rearing to cage-rearing systems can induce a robust stress response in prelaying ducks, particularly during the early stages [3]. This change can lead to sluggish growth, reduced immunity, and even mortality, ultimately impacting farming efficiency and productivity [3,4]. A study revealed that caging-induced stress not only causes liver tissue damage to some extent but also leads to an increase in the expression of inflammatory damage factors [1]. The initial 5 days following the placement of laying ducks in cages represent a period of heightened stress sensitivity [3]. During this phase, the intestinal mucosa may experience varying degrees of damage, resulting in compromised mucosal barriers, diminished immune functionality, and an imbalanced gut microbiota [5]. The subsequent 5–10 days constitute a stress recovery period, during which the intestinal tract gradually transitions from a stressed state to a normal state [6,7]. The intestinal mucosa engages in self-repair through mechanisms such as cell regeneration, immune regulation, and inflammatory resolution, collectively contributing to the restoration of intestinal barrier function [8]. In this context, investigating the mechanisms underlying intestinal stress damage and repair is particularly crucial, as this study provides a scientific foundation and guidance for the sustainable development of intensive laying duck farming.

MicroRNAs (miRNAs) are small, non-protein-coding RNAs that exist in animals, plants, and viruses [9,10]. These molecules play a key role in the posttranscriptional regulation of gene expression and affect various biological processes. Although miRNAs do not have coding potential, they can decrease or inhibit translation by targeting the 3’ untranslated region (UTR) of mRNAs to mediate gene silencing [11,12]. Thus, miRNAs can affect the occurrence of apoptosis, inflammation, cancer, and various biological processes by regulating mRNA translation [13,14]. The differential expression of miRNAs is related to intestinal oxidative stress injury and intestinal epithelial barrier dysfunction. For example, oxidative stress can induce miR-494-3p expression through the p38MAPK-c-Myc signaling pathway [15]. Overexpression of miR-21 can downregulate the expression of ROCK1, inhibit the Rho-ROCK pathway, and protect the intestinal barrier [16]. In ducks, apla-miR-217-5p may be involved in the regulation of stress-induced intestinal mucosal injury through the regulation of CHRDL1 [17]. In addition, the long noncoding RNA uc.173 can specifically stimulate the translation of the tightly associated molecule claudin-1 by binding to miR-29b, which enhances intestinal epithelial barrier function [18]. The biological importance of many miRNAs in intestinal epithelial barrier repair has been elucidated, and functional studies have suggested that these molecules play their regulatory roles; previous research has also shown that miRNAs play regulatory roles in mammalian intestinal barrier repair. Furthermore, research on the mechanism of miRNA-mediated injury repair through posttranscriptional regulation of target genes in the stressed intestinal mucosa of ducks is still limited.

MAP3K2 is a crucial cellular signaling protein. This molecule belongs to the MAP3K family of proteins [19]. The primary function of MAP3K2 is to transmit external stimuli or signals into the cell, thereby activating downstream MAPK signaling pathways. Once activated, MAP3K2 triggers a series of cellular physiological processes, including cell proliferation, differentiation, migration, and invasion [20,21]. Such miRNAs as miR-3616-3p can target MAP3K2, regulating epithelial cell apoptosis induced by heat stress [22]. Moreover, overexpression of miR-186 can suppress the expression of MAP3K2, further inhibiting cancer cell proliferation, migration, and invasion [23]. However, the mechanism of action of MAP3K2 in the repair of intestinal barrier damage in ducks is still unknown.

In the present study, high-throughput sequencing and tandem mass tagging (TMT) proteomics technology were used to identify the differentially expressed miRNAs and proteins in the intestinal tissues of prelaying juvenile ducks in the traditional breeding (TB) and cage rearing (CR) groups. Correlation analysis and dual-luciferase reporter gene assays were subsequently used to screen for molecular pathways related to intestinal barrier repair. Finally, the role of the identified molecular pathways in duck intestinal barrier repair was validated at the cellular level; the results provide molecular targets for repairing damage to the intestinal mucosa caused by the stress of transferring prelaying ducks to cage-rearing environments.

## 2. Materials and Methods

### 2.1. Ethics Statement

The methods of this study were conducted in accordance with the Guide to Laboratory Animals developed by the Ministry of Science and Technology. All animal experimental procedures were approved by the Animal Ethics Committee of the Hubei Academy of Agricultural Sciences (2021-620-000-001-021).

### 2.2. Animals and Treatments

All ducks and duck embryos in this study were obtained from the poultry breeding experimental farm at the Hubei Academy of Agricultural Sciences (Wuhan, Hubei, China). In this study, sixty-four 90-day-old Nonghu No. 2 ducks, a new breed approved by the Chinese Livestock and Poultry Breeding Approval Committee, were selected. The ducks were pre-fed for one week before the official start of the experiment, and floor rearing was carried out on the ground during the prefeeding period. The test period was 10 days, and the animals were divided into the following two groups: the traditional breeding (TB) and cage rearing (CR) groups. Each group had 4 replicates, and each replicate included 8 ducks. The ducks in each group were fed in the same external environment and feeding management conditions, and the amount of food given to each group was consistent. 

### 2.3. Sample Collection

The samples were collected on the morning of the second day after the end of the trial. Sixty-four ducks were sacrificed by carotid artery bloodletting after being anesthetized by intravenous injection of 8 mg/kg xylazine hydrochloride (X-1251, Sigma-Aldrich, Carlsbad, CA, USA). A 5 cm sample from the duodenum was taken, washed with normal saline, labeled, and placed in a phosphate-buffered saline (PBS) solution; then, the sample was transferred to liquid nitrogen for freezing and stored in a −80 °C freezer for later transcriptome sequencing.

### 2.4. Morphological Examination

Formalin-fixed samples (TB and CR) were paraffin-embedded and laterally sliced to a thickness of 5 μm. After dewaxing and dehydration, the intestine sections were stained with hematoxylin and eosin (H&E). The morphology of the intestinal mucosa was observed using a Nikon E200 microscope (Nikon, Tokyo, Japan) and analyzed using an image analysis system. The specific measurements of intestinal villus length, crypt depth, and the villus height to crypt depth ratio were obtained through digital image processing techniques. High-resolution images of the intestinal tissue sections were captured, and specialized software was used to delineate and quantify the length of individual villi and crypt depth. The villus height-to-crypt depth ratio was then calculated by dividing the measured villus height by the corresponding crypt depth for each analyzed section.

### 2.5. RNA Extraction and Sequencing

Total RNA was extracted from the intestinal tract of each duck using a TRIzol reagent kit (Invitrogen, Carlsbad, CA, USA) according to the manufacturer’s instructions. Then, the cells were treated with RQ1 DNase (Promega, Madison, WI, USA) to remove DNA. RNA quality and quantity were assessed on a NanoDrop 1000 spectrophotometer (Thermo Fisher Scientific, Waltham, MA, USA). The RNA integrity was determined by an Agilent Bioanalyzer 2100 system (Agilent Technologies, Santa Clara, CA, USA). The RNA concentration was detected by a Qubit^®^ RNA Assay Kit with a Qubit^®^ 2.0 fluorometer (Thermo Fisher Scientific, Waltham, MA, USA). The miRNA sequencing libraries were prepared using the TruSeq Small RNA Sample Preparation Kit (Illumina, San Diego, CA, USA) according to the manufacturer’s instructions. The purified library was assessed using a Qubit 3.0 fluorometer (Invitrogen), and the size distribution was analyzed using an Agilent 2200 Bioanalyzer (Agilent Technologies, Inc., Guangzhou, China). The library was subsequently on an Illumina HiSeq platform (Illumina Inc., San Diego, CA, USA). The miRNA expression levels were normalized based on the number of fragments per kilobase (FPKM) per transcript. The differentially expressed miRNAs in both groups were identified based on FPKM ≥ 100.00 and *p* < 0.01.

### 2.6. Protein Extraction and TMT Labeling

Total protein was extracted from the collected intestinal tissues using lysis buffer containing a 1% protease inhibitor cocktail. The protein concentration was determined by a bicinchoninic acid protein assay kit (Applied Biosystems, Foster City, CA, USA). Equal amounts of protein were digested using trypsin and labeled with TMT 10-plex reagents (Thermo Fisher Scientific, USA) according to the manufacturer’s instructions. TMT-labeled samples were analyzed using liquid chromatography-tandem mass spectrometry (LC-MS/MS) for protein identification and quantification. Peptide identifications with false discovery rates >1% (*p*-value > 0.01) were discarded. The protein grouping option was disabled in all analyses. For the annotation of cellular localization, data were analyzed through the use of QIAGEN Ingenuity^®^ Pathway Analysis (IPA^®^, QIAGEN, Redwood City, CA, USA, www.qiagen.com/ingenuity, accessed on 20 November 2021).

### 2.7. Quantitative Real-Time PCR

Premier 5.0 software was used to design primers according to the sequences of the miRNAs, and the reaction mixture was placed on ice according to the instructions. A LightCycler^®^ 96 system (Roche, Life Science, Baden-Württemberg, BW, Germany) was used to perform qRT-PCR with cDNA of pre-reverse transcription and the SYBR^®^ PrimeScriptTM RT-qPCR Kit (Roche, Mannheim, Germany). U6 was used as an internal reference gene, and the primer sequences are shown in Table 1. The relative expression of miRNAs was calculated by the 2^−ΔΔCt^ method.

### 2.8. Establishment of the Cell Oxidative Stress Model 

Duck intestinal epithelial cells were isolated from six No. 2 laying duck embryos at embryonic day 26 through type I collagenase (Gibco, Grand Island, NY, USA, Thermo Fisher Scientific). The cells were cultured in DMEM/F12 (Sigma-Aldrich, St. Louis, MO, USA) supplemented with 5% FBS (Mediatech Inc., Manassas, VA, USA) and incubated at 5% CO_2_ and 37 °C in 24-well cell culture dishes (Sangon, Shanghai, China) at a density of 3 × 10^5^ cells/cm^2^. After 24 h, except for those in the blank control group, the cells in each group were treated with 50 μmol/L H_2_O_2_ for 4 h to induce oxidative stress and establish a duck intestinal epithelial cell (dIEC) oxidative stress model.

### 2.9. Cell Transfection

The cells were divided into the following three groups after 24 h of culture: the blank control group (cells not subjected to oxidative stress treatment), the miR-106a-5p mimic group, the miR-106a-5p inhibitor group, and the negative control group (NC; cells subjected to oxidative stress treatment but not transfected). Transfection was performed using Lipofectamine 3000 (Invitrogen, Carlsbad, CA, USA) according to the manufacturer’s instructions. Twenty-four hours after transfection, the cells were collected to determine the expression of MAP3K2.

### 2.10. Cell Viability Determination

Cell viability was assessed using the Cell Counting Kit-8 (CCK-8) assay (BS350A, Biosharp, Labgic Technology Co., Ltd., Shanghai, China) in accordance with standard procedures. The transferred cells were digested with enzymes in 6-well plates and seeded into 96-well plates (2 × 10^3^ cells/well). Then, 20 μL of CCK-8 reagent was added to each well, and the plates were incubated for 4 h. Subsequently, the absorbance at a wavelength of 450 nm was measured using a microplate reader (Victor X5, PerkinElmer, Singapore). The cell viability percentage was calculated using the following formula: cell viability = (mean absorbance in the test wells)/(mean absorbance in the blank control wells) × 100%. Cell viability was determined after 12 h, 24 h, and 36 h of treatment.

### 2.11. Measurement of Cell Transmembrane Resistance

Transepithelial electrical resistance (TEER) was measured using an epithelial voltohm meter with a chopstick electrode (Millicell ERS-2, Millipore, Billerica, MA, USA). The cells were seeded in Transwell^®^ 6-well plates (5 × 10^4^ cells/well) (Costar 3412, Corning, Grand Island, NY, USA). The electrode was immersed at a 90° angle, with one tip in the basolateral chamber and the other in the apical chamber. The TEER was calculated as follows: TEER (Ω∙cm^2^) = (cell growth hole value-blank hole value) × micropore area. The area of the insert in the Transwell^®^ 6-well cell culture plate was 4.67 cm^2^. The TEER was measured at 0 h, 12 h, 24 h, 36 h, and 48 h after treatment.

### 2.12. Protein-MiRNA Interaction Network Construction and Pathway Analysis

The protein-miRNA interaction network was generated via a bioinformatics approach that integrated miRNA target prediction outcomes with proteomic data. Initially, potential target genes for differentially expressed miRNAs were identified utilizing established miRNA target prediction algorithms and databases. The identified target genes were subsequently cross-referenced with the differentially expressed proteins extracted from our proteomic analysis. The interactions between the selected miRNAs and their target proteins were visualized and represented as an interaction network using specialized network visualization tools. In addition, Gene Ontology (GO; http://www.geneontology.org, accessed on 20 November 2021) was used for functional analysis of these genes, and the Kyoto Encyclopedia of Genes and Genomes (KEGG; http://www.genome.jp/kegg/, accessed on 20 November 2021) was used for pathway analysis.

### 2.13. Dual-Luciferase Reporter Assay

The 3′-UTR of the miRNA-binding site containing the target gene was amplified with the pmirGLO Dual-Luciferase miRNA target expression vector according to the manufacturer’s instructions. The verified vector plasmid was transfected into primary duck intestinal epithelial cells with miRNA mimics, miRNA inhibitors, and a negative control (NC, GenePhama, Shanghai, China) by Lipofectamine 3000 (Thermo Fisher Scientific, Waltham, MA, USA). After 24 h of transfection, the luciferase activities were measured using a luciferase reporter assay (Promega, Madison, WI, USA). The oligonucleotide sequences are listed in Table 2.

### 2.14. Western Blotting

Total protein was extracted from the cell samples with an RIPA buffer solution containing the protease inhibitor bicinchoninic acid (Solarbio, Beijing, China). The proteins were separated via polyacrylamide gel electrophoresis. Next, the proteins were transferred to a polyvinylidene fluoride membrane and blocked with 5% skim milk for 2 h. After sealing, the membranes were washed with TBST 3 times for 10 min each. The membrane was incubated with the primary antibody at 4 °C overnight and then washed with TBST 3 times for 10 min each. Then, the membrane was incubated with the secondary antibody at room temperature for 2 h. The luminous solution (A solution, B solution) was added at a 1:1 ratio in the dark. A pipette was used to obtain a suitable amount of luminous solution to cover the polyvinylidene fluoride film, which was exposed on the ECL luminous instrument; then, images were collected. The antibodies used in this study include MAP3K2 Rabbit mAb (MEKK2, R24953, ZENBIO, Chengdu, China), GAPDH Rabbit mAb (A19056, ABclonal, Wuhan, China), and HRP Goat Anti-Rabbit IgG (AS014, ABclonal, Wuhan, China). GAPDH was used as a loading control. Positive control (PC) is validated mouse embryonic stem cells used to verify the specificity of the MAP3K2 antibody.

### 2.15. Statistical Analysis

Data analysis was performed using SPSS 23.0 software (IBM, Inc., Armonk, NY, USA) based on the two-tailed Student’s *t*-test. Differences were considered significant when *p*-values were less than 0.05. All the data are expressed as the means ± standard deviations (SDs).

## 3. Results

### 3.1. Body Weight and Morphological Examination

After 10 days of cage rearing, the intestinal villi in the CR group were shortened, the depth of the crypts was increased, and the ratio of villus height to crypt depth was decreased (*p* < 0.01; Figure 1). The body weight and food intake of the CR ducks were significantly lower than those of the TB ducks (*p* < 0.01; Table 3).

### 3.2. MiRNA Expression Profiles and Target Gene Functional Annotation

A total of 713 miRNAs were identified in all the intestinal samples by sequencing analysis. Most of the identified molecules overlapped, with a small portion being uniquely expressed in each group. The statistical results revealed 21 differentially expressed miRNAs in the TB group and CR group: 6 miRNAs were upregulated and 15 miRNAs were downregulated (Figure 2A). Cluster analysis indicated that the known differentially expressed miRNAs associated with TB and CR were clearly clustered into two classes, indicating that the samples had good uniformity among the three replicates (Figure 2B). To determine the regulatory functions of miRNAs specifically expressed in the CR group, we performed GO and KEGG pathway analyses of the target genes of the CR-enriched miRNAs. KEGG pathway analysis revealed that the target genes of the CR-enriched miRNAs were involved in mainly pathways such as the peroxisome, fatty acid metabolism, and oxidative phosphorylation, which are involved in the repair of intestinal barrier damage through processes such as metabolism, cellular repair, and energy synthesis (Figure 2C). The GO includes biological processes (BPs), cellular components (CCs), and molecular functions (MFs). GO terms with a *p* < 0.05 were considered significantly enriched. Figure 2D lists the top 30 GO terms for the TB vs. CR comparison. The BP functions related to immune system processes and phosphorylation were significantly enriched. Specifically, these enriched BP functions may play a crucial role in the regulation of immune responses, which are integral to the intestinal barrier repair process and contribute to an effective stress response. For the CC category, the extrinsic component of the membrane was identified as an enriched GO term. This enrichment implies a potential impact on cell membrane dynamics, which is pertinent to the structural integrity of the intestinal barrier. In the MF category, the target genes of enriched miRNAs were found to be enriched in GO terms such as protein binding, protein kinase activity, and binding. These enrichments suggest a regulatory role in protein interactions and kinase activities, which are critical components of signaling pathways associated with cellular responses to stress and barrier repair mechanisms.

### 3.3. Validation of Differentially Expressed MiRNA by qRT-PCR

We used qRT-PCR technology to confirm the differential expression of miRNAs (Figure 3) that were identified by high-throughput sequencing. The results showed that the fold change in expression of the miRNAs was different between the qRT-PCR and high-throughput sequencing results. However, the trend of variation was consistent between the two analytical methods.

### 3.4. Effect of Apla-miR-106a-5p on Cell Viability of dIECs 

Cell viability analysis was performed at 12 h, 24 h, and 36 h post-treatment. After 12 h of transfection, both the NC and miR-106a-5p inhibitor groups exhibited a notable reduction in cell viability compared to that of the blank control group (*p* < 0.05). Notably, throughout the transfection period of 12–36 h, the miR-106a-5p inhibitor group exhibited significantly greater cell viability than did the NC group (*p* < 0.05). Furthermore, at both the 24-h and 36-h time points, cell viability in the miR-106a-5p inhibitor group exhibited a significant increase compared with that in the blank control group (*p* < 0.05) (Figure 4).

### 3.5. Effect of Apla-miR-106a-5p on Transmembrane Resistance in the dIECs 

According to the results of the transendothelial electrical resistance (TEER), compared with that of the blank control group, the TEER of the dIECs after H_2_O_2_ treatment decreased significantly, while the TEER of the dIECs in the apla-miR-106a-5p inhibitors group increased gradually during the subsequent 48 h of culture. In contrast, the TEER of the dIECs in the NC group decreased first and then slowly increased during the subsequent 48 h of culture, but these TEER values were significantly lower than those of the blank control group (*p* < 0.05) (Figure 5).

### 3.6. Differential Protein Expression and Pathway Analysis

Figure 6A shows the distribution of differentially expressed proteins between the TB group and the CR group. There were 45 proteins differentially expressed between the CR and TB groups: 23 upregulated proteins and 22 downregulated proteins. Figure 6B shows the clustered heatmap of differentially expressed proteins. The results of the KEGG analysis shown in Figure 6C revealed that the differentially expressed proteins in the duodenum of the CR and TB group ducks were enriched mainly in the MAPK signaling pathway (involving mitogen-activated protein kinase 2, MAP3K2, and mitogen-activated protein kinase 11, p38-β), the PPAR signaling pathway (involving fatty acid binding protein), the adipocytokine signaling pathway (involving lipid metabolism), and tight junction/gap junction pathways (involving intercellular junctions). Figure 6D lists the top 30 GO terms of the differentially expressed proteins of the TB and CR groups. The BP functions related to proteolysis, anion transport, and the immune response were significantly enriched in these proteins. The extracellular space, F-actin capping protein complex, cytoskeletal part, and bicellular tight junction were the enriched GO terms for CC. In the MF category, the differentially expressed proteins were enriched in various GO terms, including peptidyl-dipeptidase activity, metallopeptidase activity, and hydrolase activity.

### 3.7. Prediction of Targets in the 3′-UTR of Differentially Expressed miRNAs and Proteins 

Using Cytoscape 3.5.1 software, we predicted the targeted relationships between differentially expressed miRNAs and protein-coding genes based on miRNA high-throughput sequencing and TMT protein sequencing data. We screened out miRNA-mRNA pairs with post-transcriptional regulatory mechanisms (apla-miR-106a-5p/MAP3K2) (Figure 7A). Furthermore, using the bioinformatics tool TargetScan, we successfully identified the binding site of apla-miR-106a-5p in the MAP3K2 3′-UTR (Figure 7B).

### 3.8. Validation of the Negative Regulation of miRNA/mRNA Pairs

Quantitative PCR (qPCR) was used to verify the expression of apla-miR-106a-5p and MAP3K2 in tissue samples. As depicted in Figure 8A, the expression of apla-miR-106a-5p in the CR group was significantly lower than that in the TB group (*p* < 0.05). Conversely, MAP3K2 expression did not significantly differ in the two groups (Figure 8B). Further investigation revealed a significant increase in MAP3K2 protein expression in the CR group compared to that in the TB group (*p* < 0.05) (Figure 8C,D).

As shown in Figure 8E, recombinant reporters containing the 3′-UTR MAP3K2 were cotransfected into dIECs with miR-106a-5p mimics, the miR-106a-5p inhibitor, and a negative control (NC). Our findings indicated that the apla-miR-106a-5p inhibitor significantly enhanced luciferase activity in the MAP3K2 WT vector (*p* < 0.05) compared to that in the negative control (NC), while the apla-miR-106a-5p mimic significantly decreased luciferase activity of the MAP3K2 WT vector (*p* < 0.05). Transfection of apla-miR-106a-5p had no impact on the mutant form of MAP3K2.

## 4. Discussion

Duck eggs are a good source of protein and are enriched in various vitamins and minerals, making them pivotal for human nutrition [24]. In recent years, intensive duck farming has been widely adopted as a secure, environmentally friendly, and efficient farming model [25]. This approach involves initially raising ducklings in open-ground conditions and later transitioning them to intensive farming cages as they mature for production. However, during this transition, ducks are susceptible to stress-induced intestinal barrier damage, leading to reduced farming efficiency [26,27]. An increasing body of evidence suggests that miRNAs play a crucial role in the repair of intestinal barrier damage. In this study, we found that caged stress led to a decrease in body weight and feed intake in ducks, accompanied by elongated villi and deepened crypts, impacting intestinal barrier function. Additionally, caged stress resulted in a reduction in apla-miR-106a-5p expression and an increase in MAP3K2 expression. Furthermore, apla-miR-106a-5p was found to functionally target MAP3K2, inhibiting its expression and promoting the repair of intestinal barrier function during stress conditions.

MiRNAs are endogenous small noncoding RNAs that have been shown to be involved in various cellular processes, including proliferation, differentiation, migration, invasion, and apoptosis, according to numerous studies [28,29]. Importantly, miRNAs play a pivotal role in the repair of oxidative stress damage to the intestinal barrier [30,31,32]. In our study, miRNA sequencing analysis revealed 21 differentially expressed miRNAs in the intestines of cage-reared ducks compared to those in the traditional breeding group. Among these miRNAs, miR-106a-5p is a significantly downregulated key miRNA.

MiR-106a, a member of the miR-17 family, has garnered increased attention for its abnormal regulation in the context of inflammation and oxidative stress within the body [33]. Moreover, miRNA-106a-5p has been established as a pivotal regulator of various biological processes, including cell proliferation, migration, autophagy, and ferroptosis [34,35]. A previous study revealed a significant downregulation of miR-106a-5p in an H_2_O_2_-induced cellular oxidative stress model, which is indicative of premature aging [36]. Additionally, research has highlighted the role of miR-106a-5p in alleviating apoptosis and oxidative damage in human umbilical vein endothelial cells induced by oxidized low-density lipoprotein; this effect was achieved through the targeting of STAT3 and the reduction of reactive oxygen species (ROS) levels in vivo [37]. In this study, we observed a substantial reduction in the expression level of miR-106a-5p in the intestines of cage-reared ducks during the injury repair period. Moreover, the inhibition of apla-miR-106a-5p expression increased cellular viability and reduced intestinal barrier permeability. These findings strongly suggest that miR-106a-5p plays a role in repairing intestinal mucosal damage caused by the stress associated with cage rearing in ducks. 

MAP3K2, a member of the MAP3K family, plays a pivotal role in the IL-18/MAP3K2/JNK pathway, which is essential for T-cell immunity in the intestine and shows promise as a new target for intervention in T-cell-mediated colitis [38]. Furthermore, recent research has revealed that intestinal stromal cells can regulate the proliferation and differentiation of intestinal stem cells by enhancing WNT signaling through MAP3K2. This effect on intestinal regeneration is mediated by the ROS/MAP3K2/ERK5/KLF2 axis [21]. Notably, studies have also demonstrated that the levels of MAP3K2 in serum samples from patients with acute myocardial infarction are elevated. Furthermore, miR-1184 has been identified as a regulator that targets and inhibits the expression of MAP3K2 to mitigate hypoxia-induced injury in AC16 cells [39]. Through TMT protein sequencing, we observed significant upregulation of the MAP3K2 protein in the caged stress group. In the present study, apla-miR-106a-5p was identified as promoting the repair of intestinal barrier damage by targeting MAP3K2. Additionally, the 3′-UTR region of MAP3K2 contains binding sites for apla-miR-106a-5p, demonstrating the direct targeting of MAP3K2 by apla-miR-106a-5p and post-transcriptional regulation of MAP3K2 at the protein level.

## 5. Conclusions

In summary, as shown in Figure 9, our results revealed that the caging process in prelaying ducks induces oxidative stress, which damages intestinal barrier integrity. Differentially expressed miRNAs and proteins were screened in the intestinal tissues of the traditional breeding and caged rearing groups of ducks. MiR-106a-5p, which was downregulated in the caged group, activated *MAP3K2* expression through transcriptional regulation by binding to its 3′-UTR, thus activating the MAPK-related signaling pathway and ultimately repairing intestinal barrier damage. These findings provide a basis for further study of the molecular mechanism of intestinal barrier repair in caged ducks under stress and help to clarify the oxidative stress that occurs during the caging process from a new perspective by using transcriptome and proteome sequencing data.

## Figures and Tables

**Figure 1 animals-14-01037-f001:**
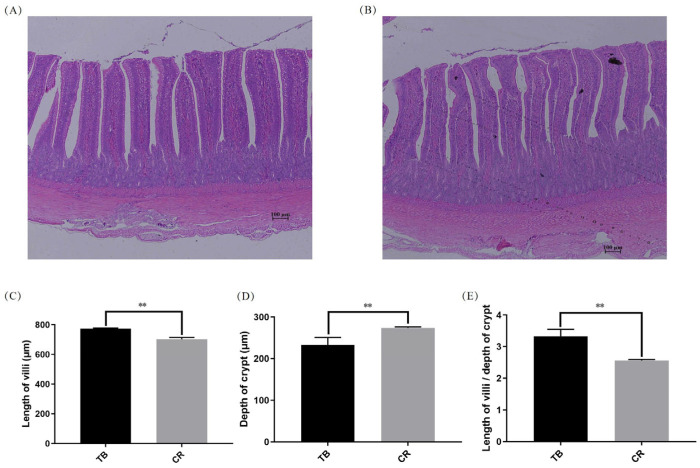
The effects of early-life stress on the structure of duodenal villi in laying ducks (40× magnification). (**A**) H&E-stained image of duodenal villi and crypts in the TB group; (**B**) H&E-stained image of duodenal villi and crypts in the CR group; (**C**) Effects of different breeding methods on the height of experimental duck villi; (**D**) Effects of different breeding methods on the depth of experimental duck crypts; (**E**) Effects of different breeding methods on the ratio of villi height to crypt depth in experimental ducks. ** indicates extremely significant differences (*p* < 0.01); Abbreviations: H&E, hematoxylin and eosin; TB, traditional breeding; CR, cage rearing.

**Figure 2 animals-14-01037-f002:**
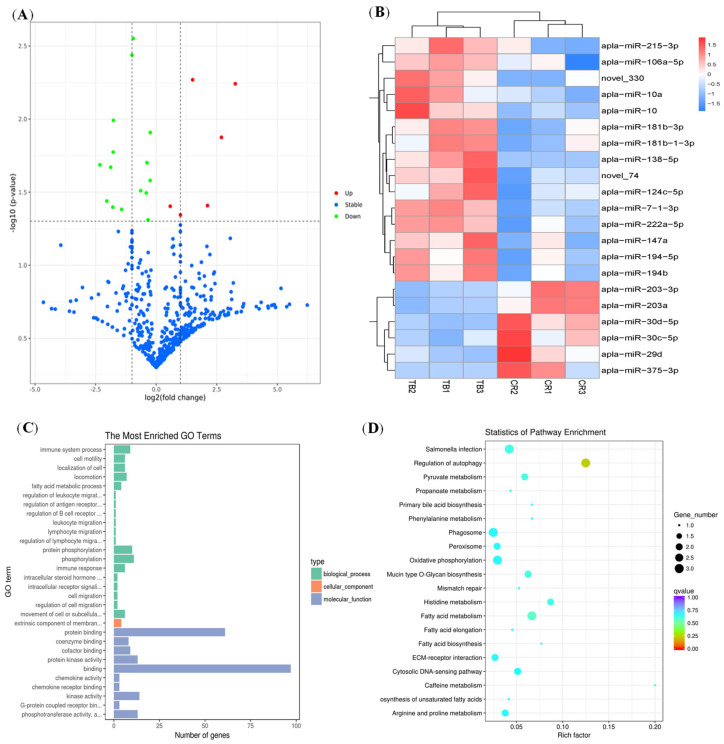
Analysis of differentially expressed miRNAs. (**A**) Volcano plot of differentially expressed miRNAs in the CR group and the TB group. The x-axis and y-axis indicate log2 (fold change) and 2log10 (FDR) of differentially expressed miRNAs in the CR group, respectively. The red color represents upregulated miRNAs, and the green color represents downregulated miRNAs. The dashed lines represent the significance thresholds in statistics. The vertical dashed line represents the threshold for −log_2_(fold change), while the horizontal dashed line represents the threshold for −log_10_(*p*-value). (**B**) Clustering map of differentially expressed miRNAs in the CR group and the TB group. The blue-to-red color range displays low to high expression levels, respectively. (**C**) Top 30 enriched GO terms of target genes of 21 differentially expressed miRNAs in the CR group compared with the TB group. (**D**) Top 20 enriched KEGG terms of target genes of 21 differentially expressed miRNAs in the CR group compared with the TB group. Abbreviations: FDR, false discovery rate; GO, gene ontology; KEGG, Kyoto Encyclopedia of Genes and Genomes.

**Figure 3 animals-14-01037-f003:**
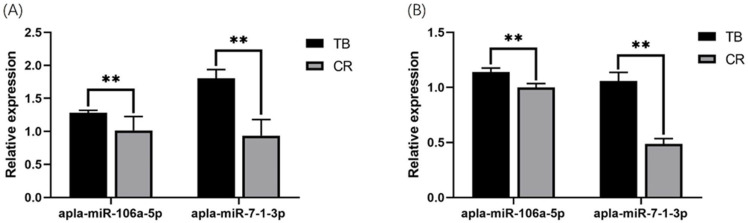
Validation of selected miRNAs by qRT-PCR. (**A**) RNA-seq of selected miRNA expression levels. (**B**) The relative expression was detected by qRT-PCR. The data are presented as the mean ± standard deviation (n = 3). Abbreviations: RNA-seq, RNA sequencing; RT-qPCR, quantitative real-time PCR. ** means *p* < 0.01.

**Figure 4 animals-14-01037-f004:**
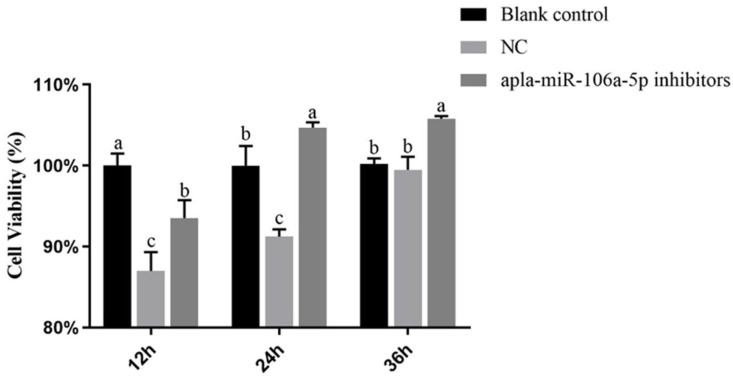
The effect of apla-miR-106a-5p on cell viability in the oxidative stress model of duck intestinal epithelial cells. NC, negative control. Different lowercase letters indicate significant differences in the apla-miR-106a-5p expression levels among the oxidative stress model in duck intestinal epithelial cells; identical letters indicate no significant difference, while different letters indicate a significant difference (*p* < 0.05).

**Figure 5 animals-14-01037-f005:**
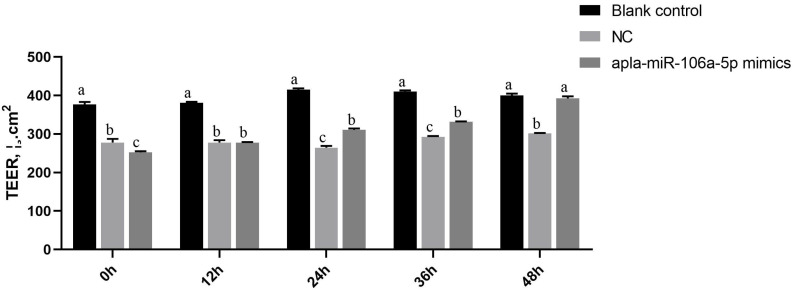
The effect of apla-miR-106a-5p on transmembrane resistance in the oxidative stress model of duck intestinal epithelial cells. NC, negative control. The lowercase letters (a, b, and c) are used to represent different levels of significance. Identical letters indicate no significant difference, while different letters indicate a significant difference (*p* < 0.05).

**Figure 6 animals-14-01037-f006:**
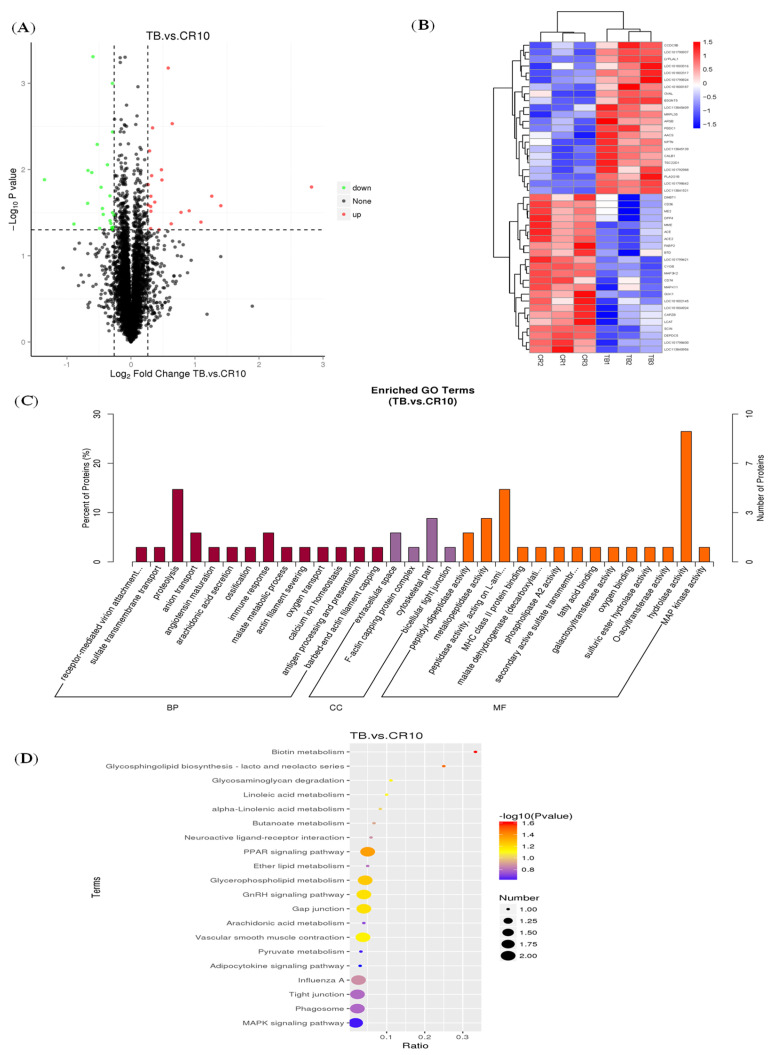
Analysis of differentially expressed proteins. (**A**) Volcano plot of differentially expressed proteins in the CR group and the TB group. The x-axis and y-axis indicate log2 (fold change) and 2log10 (FDR) of differentially expressed proteins in the CR group, respectively. The red color represents upregulated proteins, and the green color represents downregulated proteins. The dashed lines represent the significance thresholds in statistics. The vertical dashed line represents the threshold for −log_2_(fold change), while the horizontal dashed line represents the threshold for −log_10_(*p*-value). (**B**) Clustering map of differentially expressed proteins in the CR group and the TB group. The blue-to-red color range displays low to high expression levels, respectively. (**C**) Top 30 enriched GO terms of 45 differentially expressed proteins in the CR group compared with the TB group. (**D**) Top 20 enriched KEGG terms of 45 differentially expressed proteins in the CR group compared with the TB group. Abbreviations: FDR, false discovery rate; GO, gene ontology; KEGG, Kyoto Encyclopedia of Genes and Genomes.

**Figure 7 animals-14-01037-f007:**
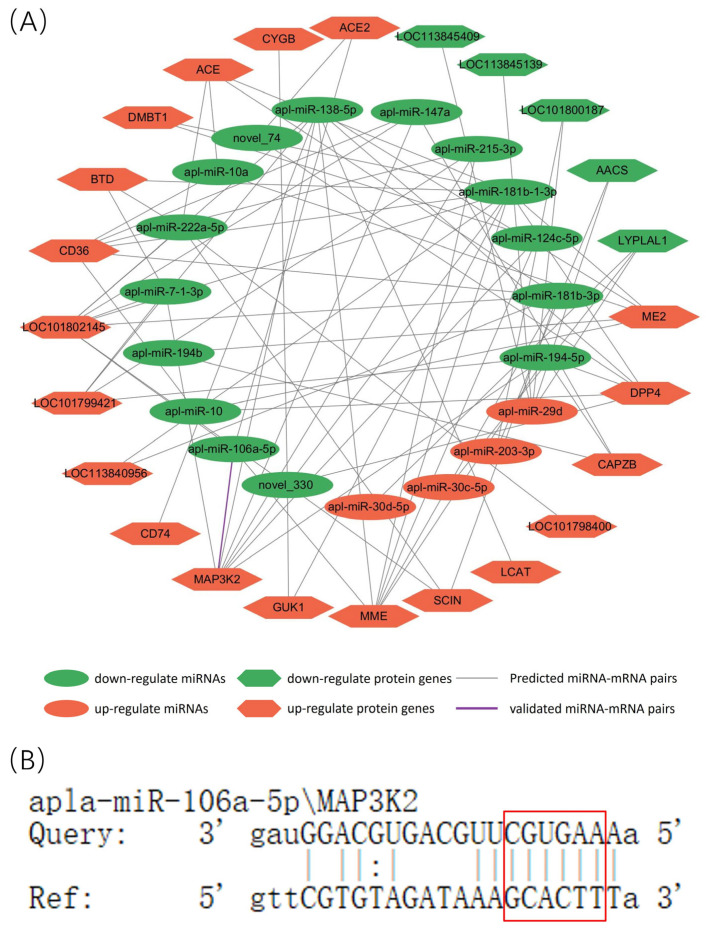
(**A**) Regulatory networks of the differentially expressed miRNAs and their putative genes involved in intestinal damage repair. (**B**) Binding sites for miR-106a-5p in the MAP3K2 3′UTR. The red box highlights the binding site for this interaction.

**Figure 8 animals-14-01037-f008:**
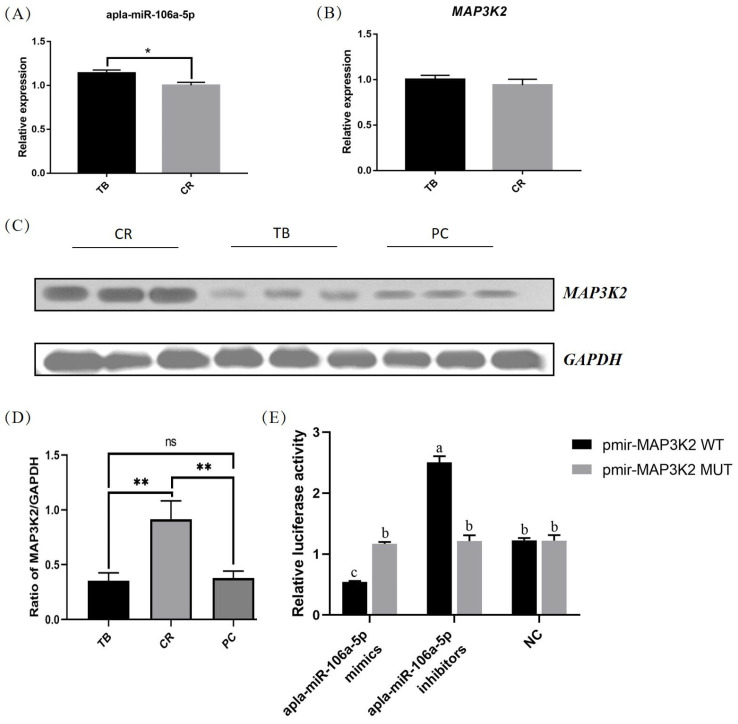
qPCR and WB validation. (**A**) Expression level of apla-miR-106a-5p in tissue samples; (**B**) Expression level of MAP3K2 in tissue samples; (**C**) Expression level of MAP3K2 protein in tissue samples; original Western blot figure is in Appendix A (**D**) Grayscale analysis of MAP3K2 protein expression level; (**E**) Dual luciferase reporter gene validation. WT, wild type; Mut, mutant; TB, traditional breeding; CR, cage rearing; PC, positive control; NC, negative control. The lowercase letters (a, b, and c) are used to represent different levels of significance. Identical letters indicate no significant difference, while different letters indicate a significant difference. (*p* < 0.05). * indicates significant difference, ** indicates extremely significant difference, ns indicates no significant difference.

**Figure 9 animals-14-01037-f009:**
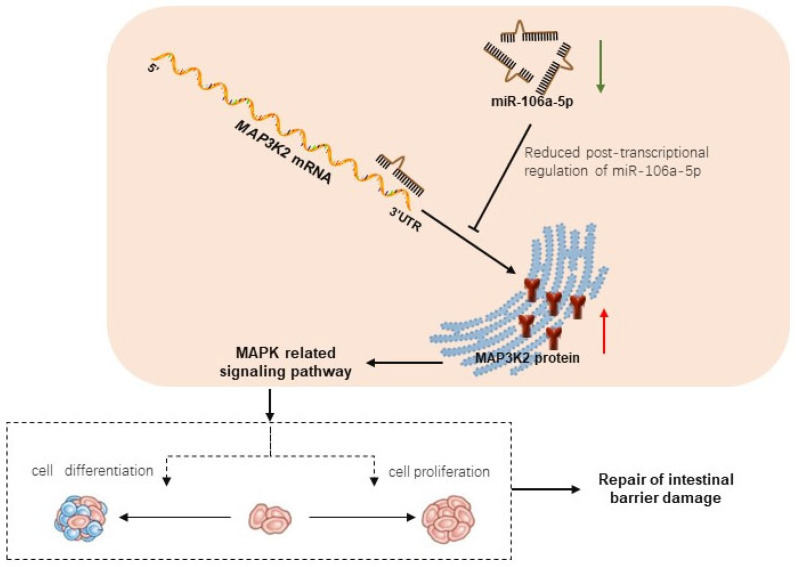
Model diagram of miR-106a-5p target *MAP3K2* promotes repair of oxidative stress damage to the intestinal barrier in prelaying ducks.

**Table 1 animals-14-01037-t001:** Primer sequences for RT-qPCR.

Primer	Sequences (5′ to 3′)
apla-miR-106a-5p-RT	CTCAACTGGTGTCGTGGAGTCGGCAATTCAGTTGAGCTACCTGC
apla-miR-106a-5p-F	CTGGTAGGAAAAGTGCTTACAGTGCA
apla-miR-106a-5p-R	TCAACTGGTGTCGTGGAGTCGGC
apla-miR-7-1-3p-RT	CTCAACTGGTGTCGTGGAGTCGGCAATTCAGTTGAG
apla-miR-7-1-3p-F	CTGGTAGGCAACAAATCACAGTCTGC
apla-miR-7-1-3p-R	TCAACTGGTGTCGTGGAGTCGGC
*MAP3K2-F*	AATACGGTGTTTGGTGTC
*MAP3K2-R*	GTGATTTGGGATAGTTGTC
U6-RT *	AACGCTTCACGAATTTGCGT
U6-F *	CTCGCTTCGGCAGCACA
U6-R *	AACGCTTCACGAATTTGCGT
*β-ACTIN-F* ^#^	ATGTCGCCCTGGATTTCG
*β-ACTIN-R* ^#^	CACAGGACTCCATACCCAAGAA

* and ^#^ indicate the housekeeping genes for miRNA and protein-coding genes, respectively. RT, stem ring primer LOOP; F, forward primer; R, reverse primer.

**Table 2 animals-14-01037-t002:** The oligonucleotide sequences.

Primer	Sequences (5′ to 3′)
apla-miR-106a-5p mimics	F: CAAAGUGCUAACAGUGCAGGUAG
	R: ACCUGCACUGUUAGCACUUUGUU
apla-miR-106a-5p inhibitor	F: CUACCUGCACUGUUAGCACUUUG
	R: ACCUGCACUGUUAGCACUUUGUU
NC mimics	F: UUCUCCGAACGUGUCACGUTT
	R: ACGUGACACGUUCGGAGAATT
NC inhibitor	F: CAGUACUUUUGUGUAGUACAA
	R: ACGUGACACGUUCGGAGAATT

NC, negative control; F, forward primer; R, reverse primer.

**Table 3 animals-14-01037-t003:** The effects of cage-induced stress on body weight gain and feed intake in laying ducks.

Item	Breeding Methods		*p*-Value
	TB	CR	
BWG(g)	49.51 ± 96.67	−42.47 ± 122.24	<0.01
FI(g)	159.38 ± 0.99	130.34 ± 4.78	<0.01

TB, traditional breeding; CR, cage rearing; BWG, body weight gain; FI, food intake.

## Data Availability

The datasets of miRNA-seq generated for this study can be found in the GEO submission database (Bioproject ID: GSE237634).

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
