# Peer review of "MiR-106a-5p by Targeting *MAP3K2* Promotes Repair of Oxidative Stress Damage to the Intestinal Barrier in Prelaying Ducks"

_animals, 2024, doi:10.3390/ani14071037_

Round 1
Reviewer 1 Report
Comments and Suggestions for Authors
The manuscript “MiR-106a-5p by targeting MAP3K2 promotes repair of oxidative stress damage to the intestinal barrier in prelaying ducks” by Zhang et al, discusses the intensive cage rearing of ducks, a widely adopted farming model, induced stress-associated intestinal barrier damage, resulting in reduced body weight and altered morphological features. The investigation revealed a pivotal role of miR-106a-5p mediated MAP3K2 in this process, highlighting their potential as molecular targets for promoting the repair of intestinal damage caused by cage rearing stress in ducks.
Overall, the study is thoroughly executed and articulated, offering comprehensive data to substantiate its findings. Specific comments below:
-
Abstract- Rephrase the sentence- MicroRNAs (miRNAs) crucial in cellular processes lack clear mechanisms in repairing oxidative stress-induced damage to duck intestinal barriers.
-
Rephrase and clump both sentences- MiRNA-seq showed significant downregulation of apla-miR-106a-5p. Inhibiting its expression enhanced cell viability and improved intestinal barrier function
-
“This can lead to sluggish 54 growth, reduced immunity, and even mortality, ultimately impacting farming efficiency 55 and productivity” It would be better to include some statistics on the duck faming globally and also the impact on it due to stress.
-
There are lot of research on the role of miR-106a-5p in regulation of oxidative stress, it would be better to add that data here to show how miR-106a-5p has been implicated in oxidative stress in several previous studies.
-
It will also be better to include any previous data on the connection between MAP3K2 and miR-106a-5p.
-
Provide appropriate reference “However, the shift from ground-based rearing to cage- rearing systems can induce a robust stress response in prelaying ducks, particularly during the early stages”.
-
Provide information on how specific measurements for intestinal villi length, crypt depth, and the villus height to crypt depth ratio were done.
-
Specify whether the 713 miRNAs detected were unique or overlapping across the TB and CR groups. This clarification can enhance the understanding of the overall miRNA landscape.
-
Elaborate on the biological significance of the GO terms and KEGG pathways enriched in the target genes of CR-enriched miRNAs. Explain how these pathways relate to intestinal barrier repair and stress response.
The article demonstrates a high standard of English and writing quality, with clear and concise language that effectively communicates complex scientific concepts. The well-structured and articulate presentation enhances the accessibility of the content, contributing to the overall readability of the study.
Author Response
Dear Reviewer,
I have included my response in the attached document. Please refer to the attached file for my detailed feedback.
Thank you for your understanding and consideration.
Best regards,
Li Zhang

Reviewer 2 Report
Comments and Suggestions for Authors
MiR-106a-5p by targeting MAP3K2 promotes repair of oxidative stress damage to the intestinal barrier in prelaying ducks
Zhang et al., Animals
The authors present a study of the effect of cage rearing on the integrity of the intestinal villi and crypt of ducks, and examine the contribution of miR-106a-5p’s regulation of MAP3K2 on the negative effects to the intestine associated with cage rearing. This is a very interesting study with important implications for the duck egg production industry, as targeted regulation of miR-106a-5p may have a significant economic impact via the maintenance of intestinal integrity and increased egg yield for commercial distribution. The conclusions are generally well supported by the data, and are not greatly overinterpreted. However, the manuscript seems to have been hastily prepared, and much of the presentation is unclear, illegible, and in some cases typographical errors are evident. In addition to the experimental suggestions below that would greatly solidify the conclusions of this study, many minor errors are noted that must be thoroughly revised prior to final publication. Several specific suggestions are listed below, but there are likely to be many other errors that were missed.
Clearly this study would benefit from a loss of function study where MAP3K2 was inhibited in intestinal cells in order to see if intestinal degradation was limited. Even more telling would be a cell viability assay during miR-106a-5p overexpression and during MAP3K2 inhibition.
The abstract and the simple summary are very similar in content. The simple summary contains a great deal of jargon that laymen may have trouble understanding. The simple summary therefore needs to be extensively revised and written in simple terms that make it easier to follow.
Lines 12 and 28, it is not clear what is meant by “economic performance”.
Line 15: The acronym TMT has not been defined. It needs to be defined before first use.
Line 60: remove the redundancy of “subsequently”
Line 93: This sentence should start with Such miRNA as miR-3616-3P…
Line 175: The format of the article needs to be changed in order to minimize white space such as that found here. Table 1 can be reduced in size in order to fit on this page, and this is recommended for greater clarity for the readers.
Figure 1 – The length markers are unnecessarily small and difficult to read even when zooming in. Please make these markers larger and clearer.
Line 301 – This should read “…to confirm…”.
Line 402 – Figure 8E needs to be referenced here.
Figures 4 and 5 – This is the first introduction of the mention of “oxidative stress model”. It is therefore entirely unclear what the statistical comparisons are being made in reference to. Furthermore, it is not clear at all what the different lower case letters are meant to represent in terms of significance. How does “a” differ from “b” and “c” in terms of significance? This must be explained in greater detail. Additionally, the figure 5 legend states that * indicates P<0.05, but there are no * on the graph. These errors must be corrected, or the data cannot be appropriately interpreted.
Figure 7 – The protein names on this chart are illegible, and the figure must be resized and the names written more clearly for proper presentation and interpretation of the data.
Figure 8E – What is the difference between the pmiR-MAP3K2 WT and Mut? This does not appear to be explained either here or in the methods, and again the lower case letters are not clearly explained here.
Line 438 – This is the first instance of discussing the “oxidative stress” model in which miR-106a-5p is downregulated. Is this the model to which the present results are being statistically compared? If so, there is little scientific rationale for this, as the relative luciferase activity levels in figure 8 cannot be directly compared to the levels from another study without raw values from that study. If a different comparison is being made from what I have described here, then this needs to be explained within the text, as it is entirely unclear.
Comments on the Quality of English LanguageMinor English grammar edits are required.
Author Response

(The authors gave the same response as above.)

Round 2
Reviewer 2 Report
Comments and Suggestions for Authors
The authors have sufficiently clarified many of the problematic issues from the original submission. The gene names in Figure 7A are still somewhat difficult to read and the alignment of some of the figures could still be improved. Publication is recommended following minor modifications in data presentation to enhance the attractiveness of the figures.
Author Response
Dear Reviewer,
Thank you for your meticulous review of our submitted manuscript and for providing valuable feedback. We have addressed the issues raised during the review process, specifically improving the readability of gene names in Figure 7A and further optimizing the arrangement of all figures to enhance overall visual appeal. We trust that these adjustments meet your expectations, contributing to a more compelling presentation of the data in our paper.
We appreciate your thorough review and guidance.
Sincerely
